# Protective Effect of Probiotics Isolated from Traditional Fermented Tea Leaves (Miang) from Northern Thailand and Role of Synbiotics in Ameliorating Experimental Ulcerative Colitis in Mice

**DOI:** 10.3390/nu14010227

**Published:** 2022-01-05

**Authors:** Napapan Kangwan, Sarawut Kongkarnka, Nitsara Boonkerd, Kridsada Unban, Kalidas Shetty, Chartchai Khanongnuch

**Affiliations:** 1Division of Physiology, School of Medical Sciences, University of Phayao, Phayao 56000, Thailand; 2Department of Pathology, Faculty of Medicine, Chiang Mai University, Chiang Mai 50200, Thailand; srawutzi@gmail.com; 3Division of Microbiology, School of Medical Sciences, University of Phayao, Phayao 56000, Thailand; nitsara_nu@hotmail.com; 4Division of Biotechnology, School of Agro-Industry, Chiang Mai University, Chiang Mai 50100, Thailand; kridsada_u@cmu.ac.th; 5Global Institute of Food Security and International Agriculture (GIFSIA), Department of Plant Sciences, North Dakota State University, Fargo, ND 58108, USA; kalidas.shetty@ndsu.edu; 6Research Center of Multidisciplinary Approaches to Miang, Science and Technology Research Institute, Chiang Mai University, Chiang Mai 50200, Thailand

**Keywords:** probiotics, synbiotics, inflammatory bowel disease, *Lactobacillus*, colitis

## Abstract

This study aimed to investigate the protective effect of probiotics and synbiotics from traditional Thai fermented tea leaves (Miang) on dextran sulfate sodium (DSS)-induced colitis in mice, in comparison to sulfasalazine. C57BL/6 mice were treated with probiotics *L. pentosus* A14-6, CMY46 and synbiotics, *L. pentosus* A14-6 combined with XOS, and *L. pentosus* CMY46 combined with GOS for 21 days. Colitis was induced with 2% DSS administration for seven days during the last seven days of the experimental period. The positive group was treated with sulfasalazine. At the end of the experiment, clinical symptoms, pathohistological changes, intestinal barrier integrity, and inflammatory markers were analyzed. The probiotics and synbiotics from Miang ameliorated DSS-induced colitis by protecting body weight loss, decreasing disease activity index, restoring the colon length, and reducing pathohistological damages. Furthermore, treatment with probiotics and synbiotics improved intestinal barrier integrity, accompanied by lowing colonic and systemic inflammation. In addition, synbiotics CMY46 combined with GOS remarkedly elevated the expression of IL-10. These results suggested that synbiotics isolated from Miang had more effectiveness than sulfasalazine. Thereby, they could represent a novel potential natural agent against colonic inflammation.

## 1. Introduction

Ulcerative colitis (UC) is one of the most common types of inflammatory bowel disease (IBD) and is characterized by chronic and recurrent inflammatory conditions of the colon and rectum [1]. The incidence and prevalence of UC are increasing worldwide, resulting in high morbidity and increased socioeconomic burdens [2]. The pathogenesis of UC involves interplay of multifactorial factors, including genetic susceptibility, environmental factors, and gut microbiota, triggering the dysregulation of gut barrier function and immune-related inflammation response [3]. The colonic damages in UC patients are driven by the immune cells and cytokines response. The progression of UC has indicated that proinflammatory cytokines including interleukin (IL)-1, IL-6, IL-9, IL-13, IL-33, and tumor necrosis factor-α (TNF-α) are usually upregulated, whereas anti-inflammatory cytokines including transforming growth factor-β (TGF-β), IL-10, and IL-37 are downregulated [4]. Intestinal barrier disruption during mucosal inflammation is involved in impairing structure and remodeling of apical junctions, leading to increase gut permeability and systemic inflammation-related diseases [3]. Unfortunately, the conventional therapeutic drugs for UC, such as anti-inflammatory, antibiotics, and immunosuppressant drugs have adverse side effects and are limited in their benefits [5]. Therefore, there is a need to explore and develop novel alternative therapeutics to reduce and counter complications and the negative symptoms from medications in UC patients. 

Currently, probiotics, prebiotics, and synbiotics are increasingly becoming effective and novel strategies for advancing therapeutics for gastrointestinal diseases. Recent investigations have reported that the administration of probiotics, prebiotics, and synbiotics effectively improved the clinical outcome of UC patients [6,7]. Probiotics are live microorganisms and provide health benefits to the host with less toxicity when received in appropriate amounts. The supplementation with probiotics has been reported to alleviate gut inflammation by modulating gut microbiota, reinforcing intestinal barrier integrity, and inhibiting inflammatory responses in animals [8,9,10,11]. In addition, clinical studies have demonstrated that oral administration of probiotics or synbiotics remarkably decreased inflammatory markers C-reactive protein (CRP) and TNF-α [12]. Recently, lactic acid bacteria (LAB), particularly *bactobacilli* and *bifidobacteria*, are the most common commercial probiotic bacteria that improved gut microbiota functions and positively affected inflammation-related diseases [13]. Furthermore, prebiotics are nondigestible oligosaccharides, including xylooligosaccharides (XOS) and galactooligosaccharides (GOS), which are selectively fermented in the intestine, enhanced beneficial bacteria function and regulated gut microbiota [14]. Prebiotic supplementation has a therapeutic effect on GI diseases [15,16]. Though controversial, other studies have reported that oral administration with GOS alone increased beneficial microbial *bifidobacteria* in the colon but did not attenuate the severity of colitis induced by 2,4,6-trinitrobenzenesulfonic acid (TNBS) [17]. Dietary XOS alone has shown that *bifidobacteria* and *lactobacilli* in the colon were increased, whereas it had a limited effect on gut barrier function in rats [18]. Currently, synbiotic therapy has been demonstrated to effectively maintain intestinal homeostasis by producing synergistic impacts and by combining probiotic and prebiotic treatments; it is expected to have a therapeutic effect against gut dysbiosis-related inflammatory diseases [19]. Similarly, synbiotic supplementation of probiotic *Bacillus coagulans* MTCC5856 spores combined with prebiotic green banana resistant starch attenuated DSS-induced gut inflammation by improving expressions of tight junction proteins, reducing CRP and IL-1β, and upregulating IL-10 level [20].

Miang is a traditional fermented food product from Assam tea (*Camellia sinensis* var. *assamica*) leaves with mixed microorganisms such as LAB and yeast. Miang is produced and consumed as a snack by the local communities of northern Thailand [21]. There are various bioactive compounds in Miang relevant to antioxidant and antimicrobial activities [22]. Previously, *L. plantarum* A9-2, *L. pentosus* A14-6, *L. pentosus* A26-8, *L. pentosus* CMY46, and *Pediococcus pentosaceus* CMY46 have been selected as the high potential probiotics among 133 LAB isolated from Miang samples collected from northern Thailand [23]. In addition, these five strains demonstrated a high survival rate (>90%) when challenged by the simulated-gastrointestinal conditions and were also susceptible to antibiotics. Among these five strains, *L. pentosus* A14-6 and CMY46 have shown high potential adhesion of probiotic bacteria to the host’s intestinal mucosa, indicated by cell surface hydrophobicity property and auto-aggregation activities. Furthermore, XOS and GOS were revealed to be highly effective prebiotic for the growth of *L. pentosus* A14-6 and CMY46, respectively [24]. Therefore, this study used *L. pentosus* A14-6 combined with XOS and *L. pentosus* CMY46 combined with GOS as the synbiotics for *in vivo* investigation. The previous research has shown that treatment with *L. pentosus* had a significant increase in beneficial *Akkermansia*, which correlated to the production of metabolites such as indolepyruvate, and pantothenic acid levels in the gut, thereby alleviating dextran sulfate sodium (DSS)-induced colonic inflammation [25]. Other studies have reported that *L. pentosus* enhanced the expression of TLR inhibitors Tollip, A20, SIGIRR, and IRAKM, resulting in downregulation of downstream MAPK and NF-κB signaling pathways on intestinal epithelial cells [26]. However, the involvement of the *L. pentosus* strain from Miang in DSS-induced intestinal inflammation remains unknown. Therefore, this study investigated the protective effect of probiotics *L. pentosus* A14-6 and *L. pentosus* CMY46, isolated from Miang, and synbiotics *L. pentosus* A14-6 combination with XOS and *L. pentosus* CMY46 combination with GOS as well as compared to an anti-inflammatory drug, sulfasalazine on DSS-induced ulcerative colitis in mice. We hypothesized that the oral administration of probiotics and synbiotics from Miang would attenuate DSS-induced intestinal inflammation by enhancing gut barrier function and reducing the production of inflammatory cytokines.

## 2. Materials and Methods

### 2.1. Reagents

Dextran sodium sulfate (DSS-molecular weight 36,000–50,000 Da, colitis grade) was obtained from MP Biomedicals (Salon, OH, USA). Sulfasalazine (purity > 98%) was purchased from Sigma Aldrich (St. Louis, MO, USA). All other reagents used were of high quality and analytical grade.

### 2.2. Preparation of Probiotics and Synbiotics

Two selected lactic acid bacteria isolated from Miang samples *L. pentosus* A14-6 and *L. pentosus* CMY46 [24] were used as probiotics in this experiment. The bacterial strains maintained in 25% glycerol stock at –80 °C were activated by culturing in MRS broth and streaked on MRS agar. The single colony of probiotics was transferred to 200 mL of sterile mL MRS broth (HiMedia, Mumbai, India) and cultivated at 37 °C for 12–18 h to reach a stationary phase. The bacterial cells in culture were harvested by centrifugation at 6000× *g* for 5 min at 4 °C, and the bacterial pellet was washed two times and resuspended with 0.01 M phosphate buffer-saline pH 7.0. The viable cell number of probiotic bacteria was enumerated on MRS agar by drop plate methods as described previously [27]. Commercial xylooligosaccharide, XOS and galactooligosaccharide, GOS (Wako, Japan) were dissolved in PBS to achieve the concentration of 1.5% (*w*/*v*). The synbiotics were prepared by mixing in a ratio of probiotics: prebiotic (1:20 by volume). The synbiotic mixture was prepared by mixing an equal amount (1 mL of each) of the probiotic and prebiotic. The *L. pentosus* A14-6 combined with XOS and *L. pentosus* CMY46 combined with GOS were freshly prepared probiotics and synbiotics.

### 2.3. Animals and Ethics Statement

The study procedures were approved by the Institutional Animal Ethics Committee, University of Phayao, Thailand (code number 610104011). All mice were handled in accordance with international guidelines for the care and use of laboratory animals. Forty-nine male C57BL/6 mice (Six-week-old) were purchased from the Nomura company (Bangkok, Thailand). Mice were housed in polyethylene cages under specific-pathogen-free conditions and maintained the temperature at 25 ± 1 °C, humidity 50 ± 10%, and 12 h light/dark cycle. The mice were for acclimatization for seven days before experimental initiation. 

### 2.4. Experimental Design 

All mice were allocated into seven groups (seven mice in each group) as follows; (1) normal group; (2) DSS group; (3) probiotic *L. pentosus* A14-6 group; (4) probiotic *L. pentosus* CMY46 group; (5) synbiotic *L. pentosus* A14-6 plus XOS group; (6) synbiotic *L. pentosus* CMY46 plus GOS group; and (7) sulfasalazine group. The mice were orally gavaged daily with probiotics or synbiotics for 21 days, at a concentration of 1 × 10^9^ CFU/200 μL/day. Acute colitis was then induced by administering 2% DSS in drinking water during the last seven days of the experimental period of all groups except for the normal group that received drinking water. The positive group was treated with sulfasalazine (50 mg/kg/day) during DSS induction. The model for DSS-induced ulcerative colitis in the present study was established based on previous studies [20,28]. The dose of probiotics and sulfasalazine was selectively based on previous literature [29,30]. The probiotic and synbiotic were prepared in a vehicle of PBS and fresh each day. Sulfasalazine was dissolved in 0.5% carboxymethyl cellulose. The experimental design in this study is present in Figure 1.

### 2.5. Sample Collection

All mice were humanely euthanized by CO_2_ asphyxiation after DSS induction for seven days. Blood samples were immediately collected from cardiac puncture and centrifuged at 3000× *g* for 15 min. The spleen was carefully removed, rinsed, and weighed. The colon was harvested, opened longitudinally, and gently cleaned with ice-cold PBS. The length of the colon was measured and photographed. The distal colonic tissue was isolated for histological evaluation. A piece of the colon was kept in RNAlater device for protection of degradation until the later desired time of RNA extraction.

### 2.6. Assessment of Colitis Severity

The severity of colitis was evaluated using disease activity index (DAI) scoring. DAI was assessed on the clinical parameters and scored from the sum of body weight loss (0, none; 1, 1.0–5.0%; 2, 1.0–5.0%; 3, 10.1–15.0%; 4, >15.0%), stool consistency (0, Normal; 2, Soft with normal form; 4, loss of form/diarrhea), and fecal bleeding (0, normal; 2, detectable blood with occult test; 4, visible blood without test) [31,32]. A blinded observer performed DAI scoring. 

### 2.7. Histopathologic Analysis

The distal colon section (~0.5 cm) was kept in 10% buffered formalin solution overnight prior to embedding in paraffin. The paraffin sections were subsequently stained with hematoxylin and eosin (H and E). Blind review by gastrointestinal specialists analyzed the histopathological index. The scoring of histological damage was graded in each colon section as described in previous study, following previous published methods of Kangwan et al. [33]; (a) severity of ulceration/erosion (0, epithelium intact; 1, involvement of the lamina propria; 2, involvement of the submucosa; 3, into colon wall); (b) the area affected by intestinal inflammation (0, none; 1, <10%; 2, 10%; 3, <10–50%; 4, >50%); (c) extension of follicle aggregate (0, none; 1, mild; 2, moderate; and 3, severe); (d) edema (0, none; 1, mild; 2, moderate; and 3, severe); (e) crypt loss (0, none; 1, <10%; 2, 10%; 3, <10–50%; 4, >50%); (f) the infiltration of inflammatory cells (0, none; 1, mild; 2, moderate; and 3, severe). 

### 2.8. Determination of Goblet Cells 

Mucus-secreting goblet cells in the distal colon section were assessed using a commercial periodic acid-Schiff (PAS) staining kit (Ventana Medical System, Tucson, AZ, USA) according to the manufacturer’s instructions. The PAS staining was scored as excellent-to-poor based on preservation of mucus production (0–10), as described in the previous study [34].

### 2.9. Reverse Transcription-Quantitative Polymerase Chain Reaction (qRT-PCR)

Total RNA was prepared from the colonic tissue using TRIZol® reagent (Invitrogen; Carlsbad, CA, USA). The cDNA was generated by reverse transcription using a ReverTraAce® RT-qPCR Kit (TOYOBO, Tokyo, Japan). The synthesized cDNA was consequently amplified by qPCR using the SensiFAST SYBR® Lo-ROX Kit (Bioline, Singapore). GAPDH gene was used as the reference gene. The fold change of the genes was calculated by the method of 2^-ΔΔCt^. The primer sequences are presented in Table 1.

### 2.10. Measurement of Proinflammatory Cytokines in Plasma

Proinflammatory cytokines of the plasma, including TNF-α, IL-1β, and IL-6, were detected by the commercial ELISA kits (BioLegend, San Diego, CA, USA) as described by the manufacturer.

### 2.11. Statistical Analysis

Data were presented in term of the means ± standard error of the mean (SEM) for each group. Multiple comparisons were assessed using the one-way ANOVA, followed by Tukey’s post hoc test for comparative analysis. The % body weight loss was analyzed using the two-way analysis of variance (ANOVA), followed by Tukey’s post hoc test. Differences were considered statistically significant at a *p*-value < 0.05.

## 3. Results

### 3.1. Probiotics and Synbiotics from Miang Attenuated Clinical Symptoms and Severity of Colitis Induced by DSS

All treatments of probiotic and synbiotic for 21 days which were started 14 days before DSS administration did not result in any sign of toxicity. This was evaluated based on the mice body weight and general appearance. Mice received 2% DSS administration in drinking water for seven days continuously after day fourteen, resulting in colonic inflammation as demonstrated by body weight loss and a high score of DAI compared to the normal group (Figure 2a,b). A significant gradual decrease in body weight loss in the DSS group was detected on the fifth to the seventh day of the DSS induction period. However, all treatments with probiotic A14-6, CMY46, synbiotic A14-6 plus XOS, synbiotic CMY46 plus GOS, and sulfasalazine effectively attenuated DSS-induced body weight loss on day seven (Figure 2a). In addition, all treatments of probiotics and synbiotics, but not sulfasalazine treatment, significantly reduced DAI score (Figure 2b) as reflected by the reduction of weight loss score (Figure 2c), lower incidences of diarrhea, and decreased fecal bleeding score (Figure 2e) without altering stool consistency score (Figure 2d). Sulfasalazine did not reduce the DAI score in the present study.

Colon shortening is a typical feature of colitis induced by DSS [35]. All treatment of probiotics and synbiotics remarkably restored the shortening of the colon compared to the DSS group (Figure 2f,g). Additionally, DSS-induced colonic inflammation, associating with enlargement of the spleen, was prevented by all of the treatments of probiotics and synbiotics (Figure 2h). Interestingly, the effect of all treatment of probiotics and synbiotics on preservation colon length and spleen weight were equal to the sulfasalazine treatment effects. These results revealed that all treatments with probiotics and synbiotics attenuated the clinical symptoms and severity of colitis in DSS-treated mice.

### 3.2. Effects of Probiotics and Synbiotics from Miang on the Histological Damages in DSS-Induced Colonic Inflammation

DSS induced acute inflammation and histological damages in the colon, as demonstrated by increasing crypt distortion, submucosal edema, and inflammatory cellular infiltration in the colon, leading to mucosal destruction (Figure 3). Thus, the DSS group had a high histopathological index than the normal group. Treatment with synbiotics A14-6 plus XOS and synbiotic CMY46 plus GOS, but not in the probiotic alone, prevented DSS-induced histological colonic damage, as evidenced by maintaining mucosal structure, reducing crypt distortion and submucosal edema, and decreasing the infiltration of inflammatory cells (thereby lowering the histopathological index). Noticeably, treatment with CMY46 plus GOS (*p* < 0.01) was more effective than sulfasalazine treatment (Figure 3). 

### 3.3. Probiotics and Synbiotics from Miang Improved the Colonic Barrier Integrity in DSS-Induced Colitis

The presence of mucus secreted by goblet cells in the colon is essential for the gut defense mechanisms of the host. Thus, the mucus-secreting goblet cells in the colon were assessed using periodic acid-Schiff (PAS) staining. The mucus-secreting goblet cells were observed mainly in the normal group but were lost in the DSS group, as indicated by the PAS staining score. A significantly high PAS staining score was found in colon sections of mice treated with probiotic A14-6 and synbiotic A14-6 plus XOS and synbiotic CMY46 plus GOS despite DSS induction. Treatment with synbiotic CMY46 plus GOS had a protective effect on goblet cells destroyed by DSS than sulfasalazine treatment (Figure 4).

Next, we investigated the involvement of probiotics and synbiotics on mucosal barrier integrity in the colon. The expression of intestinal epithelial cell tight junction (ZO-1, occluding, and claudin-1) and cell surface mucin (MUC-1) were also determined in the colonic tissue by qRT-PCR. As shown in Figure 5a–d, our results indicated that ZO-1 was significantly downregulated but did not involve the other tight junctions challenged by DSS. Interestingly, probiotic A14-6, synbiotic A14-6 plus XOS, and synbiotic CMY46 plus GOS remarkably improved the tight junction patterns (ZO-1, occludin, claudin-1, and MUC-1) similar to that of sulfasalazine treatment. The effect of probiotic CMY46 was less noticeable for claudin-1. Thus, treatments of synbiotic A14-6 plus XOS and synbiotic CMY46 plus GOS completely improved the loss of all tight junction proteins in DSS-induced intestinal barrier dysfunction.

### 3.4. Probiotics and Synbiotics from Miang Alleviated Expression of Colonic Inflammatory Mediators in DSS-Induced Colitis

Several inflammatory mediators and enzymes are involved in the progression of UC [36]. The anti-inflammatory effect of probiotics and synbiotics on DSS-induced inflammatory responses was measured in the colon using qRT-PCR analysis. The expression levels of inflammatory cytokines and enzymes, including TNF-α, IL-1β, IL-6, and COX-2 in the colon were markedly elevated. Meanwhile, anti-inflammatory cytokine IL-10 was significantly reduced in the DSS group compared to the normal group (Figure 6a–e). 

By contrast, treatments with probiotics and synbiotics significantly suppressed the expression of TNF-α and IL-6, similar to the effect of sulfasalazine treatment. A significant decrease in the expression of IL-1β was observed in the treatment of probiotic A14-6 and synbiotic CMY46 plus GOS. Additionally, COX-2, an enzyme responsible for local inflammation was effectively downregulated in response to treatment with probiotics CMY46, synbiotic A14-6 plus XOS, and synbiotic CMY46 plus GOS compared to the DSS group. There was no significant difference in the treatment of probiotic A14-6 and sulfasalazine compared to the DSS group. The only treatment which significantly upregulated the expression of IL-10 in the DSS group was synbiotic CMY46 plus GOS. These results confirmed the anti-inflammatory effect of probiotic and synbiotic administration, and synbiotic CMY46 plus GOS had the highest efficacy.

### 3.5. Probiotics and Synbiotics from Miang Attenuated Systemic Inflammation Induced by DSS

As shown in Figure 7a–c, serum levels of the TNF-α, IL-1β, IL-6 effectively increased in the DSS group compared to the normal group. By contrast, treatment of probiotic CYM46 and synbiotics markedly prevented the onset of systemic inflammation equal to the effect of sulfasalazine treatment. Treatment of probiotic A14-6 notably reduced serum levels of IL-1β.

## 4. Discussion

This study has demonstrated the protective effect against colonic inflammation induced by DSS by novel strains of *L. pentosus* A14-6, CMY46, which was isolated from traditional fermented tea leaves (Miang) in Northern Thailand. The benefits were also observed in response to synbiotic *L. pentosus* A14-6 plus XOS and synbiotic *L. pentosus* CMY46 plus GOS. Our findings show that the prolonged treatment of probiotics and synbiotics isolated from Miang in mice does not cause mortality. DSS induced colonic inflammation, which displayed severe clinical manifestations, histological change, destruction of intestinal barrier integrity, and increased intestinal inflammation, leading to systemic inflammation. Treatment of probiotics and synbiotics efficiently protected colonic inflammation by attenuating clinical and histological damage features, improving the integrity of the intestinal barrier, alleviating local and systemic inflammation induced by DSS. These findings suggest that the synbiotic *L. pentosus* A14-6 plus XOS, synbiotic *L. pentosus* CMY46 plus GOS provided protective effects more than sulfasalazine. Thus, synbiotics from Miang exerted a therapeutic effect in ameliorating colonic inflammation induced by DSS. The proposed mechanism of treatment response to probiotics and synbiotic isolated from Miang leading to amelioration in DSS-induced colitis is showed in Figure 8.

DSS-induced colitis in animal models are beneficial and useful to investigate the pathogenesis of colitis, factors involving colitis, and novel therapeutic strategies for inflammation-associated GI diseases such as IBD and colitis-associated carcinogenesis [35]. When DSS reaches to gut luminal, it acts as a chemical toxin to destroy the integrity of the colonic mucosal barrier leading to increased gut permeability and aggravated inflammatory responses in local and systemic inflammation [37]. Clinical and histopathological characteristics of DSS-induced colitis reflect those seen in human IBD [38]. In this study, severe clinical symptoms of colitis were observed in DSS-treated mice for seven days compared to normal mice, including body weight loss, diarrhea, and bloody stools leading to the high value of DAI, shorting of colon length as well as spleen enlargement (a marker of severity in DSS-induced colitis). In addition, the histological damages of DSS-induced colitis, including epithelial destruction, crypt distortion, submucosal edema, and neutrophils infiltration of lamina propria and submucosa was observed in DSS-treated mice. Treatments with probiotics, synbiotics, and sulfasalazine for 21 days, had attenuated these clinical symptoms and attenuated colitis. However, sulfasalazine did not decrease the DAI. Also, this study demonstrated that only treatment with synbiotics remarkably reduced histological damage in the colon. These findings are consistent with previous studies. Sulfasalazine has been used widely as the first-line of anti-inflammatory drug therapeutics for UC therapy. It is associated with severe adverse effects after long-term and high-dose usage [39]. A previous study has shown that probiotics mixtures (with *L. reuteri*, *B. coagulans*, *Bifidobacterium longum*, and *Clostridium butyricum*) were more effective than sulfasalazine to alleviate DSS-induced colitis [11]. Sulfasalazine treatment did not prevent weight loss, reduce DAI, and histological damage in mice that received DSS. Other studies have shown that *L. plantarum* 06CC2 strain, isolated from traditional Mongolian dairy products, prevented DSS-induced colitis in mice by attenuating the DAI score, pathologic index, and an anti-inflammatory response [40]. In addition, supplementation of synbiotic containing *B. infantis* and XOS reduced the DAI and histopathological scores. Synbiotic was more effective than the probiotic or prebiotic alone [26]. A clinical study has shown that treatment with synbiotic containing *B. longum* (2 × 10^9^ CFU/day) with psyllium (8 g/day) could reduce clinical symptoms and improve the quality of life of UC patients. Hence, synbiotic therapy has been observed to be more effective than either the treatment with probiotic or prebiotic [41]. 

Alternative colonic mucus properties and disruption of the integrity of gut barrier are vital roles in the pathogenesis of UC and animal models show that this further promotes dysregulated immune response, leading to aggravating inflammation of the intestinal mucosa [1]. UC is associated with mucus-producing defects leading to diminished number of goblet cells [42]. Our study demonstrated that 2% DSS induction disturbed mucosa barrier function as indicated by reduced mucus-secreting goblet cells in colonic mucosa and leading to downregulation of tight junction, especially ZO-1. However, other investigators showed that mice treated with 3% DSS exposure for seven days induced mucosal barrier dysfunction. Similarly, 2.5% DSS induction could downregulate tight junctions, including ZO-1, occludin, and claudin-1 [11]. Therefore, disturbance in mucosal barrier function depended on the concentration and duration of DSS administration. In this study, all treatments with probiotic, synbiotic, and sulfasalazine have shown improved mucosal barrier integrity as reflected by an increasing number of goblet cells and upregulated tight junction ZO-1, occludin, claudin-1 as well as MUC1, cell surface mucin in mucosal tissues [36]. Probiotics and synbiotics treatments have shown that improving the gut barrier function as one of the crucial factors to alleviate the symptoms of UC [11,25]. Surprisingly even treatment with *L. reuteri* alone prevented colitis by reducing the translocation of bacteria from the intestinal mucosa into the mesenteric lymph nodes but not improving the mucus layer integrity or the mucus microbiota induced by DSS [43]. Therefore, the probiotic and synbiotic use in the present study could effectively protect DSS-induced colitis by enhancing mucosal barrier function. However, preventing colitis depended on probiotic strains, types of prebiotic that synergize the probiotic activity.

The mucosal barrier dysfunction in UC also leads to dysregulated immune responses resulting in inflammation and tissue destruction [44]. Our results indicated that the treatment of probiotics and synbiotic has excellent immunomodulatory and anti-inflammatory efficacy, as evidenced by the reduction in colonic inflammatory mediator expressions of TNF-α IL-1β, Il-6, and COX-2. Moreover, IL-10 is an essential anti-inflammatory cytokine and ameliorates the overproduction of proinflammatory cytokines [41]. It was elevated in only in response to treatment with synbiotics CMY46 plus GOS. Synbiotics have been shown more effective than probiotics alone since they can selectively stimulate bacteria in the gut. However, probiotics have to compete with the normal gut microbiota and large amounts of these probiotics to transit and survival through the gastrointestinal tract [45].

Beyond the anti-inflammatory activity in the colon, the probiotic and synbiotic have the potential for alleviation of DSS-induced systemic inflammation. Evidence has shown that increased gut permeability causes luminal microbiota and bacterial toxins to spread across systemic circulation, leading to systemic inflammation. Also, DSS-induced colitis increased systemic inflammation, as demonstrated by increasing IL-6 in serum level and elevated levels of IL-6 and TNF-α levels in cortical tissue, which then results in cortical inflammation. Studies have also found activation of microglial cells and decrement in occludin and claudin-5 expression in the brain tissue after DSS-induced colitis [46]. Therefore, probiotics, prebiotics, and synbiotics have been proven to be effective strategies to prevent and relieve UC through gut microbiota modulation. In this study, probiotic, synbiotic, and sulfasalazine treatments suppressed serum levels of TNF-α, IL-1β, and Il-6. However, the beneficial effect of probiotics and synbiotics on DSS-induced colitis seen in this study could be due to the modulation of gut microbiota composition which was not investigated in this study. Hence, further study is required to assess the effect of probiotics and synbiotics from Miang on gut microbiota and their metabolites in an animal model and clinical. Overall findings of this study are consistent with the previous study which demonstrated that supplementation heat-killed *L. pentosus* S-PT84 exerted anti-inflammatory activity in lipopolysaccharides-induced low-grade chronic inflammation associated metabolic disorders in mice via maintaining tight-junction protein expression, reducing infiltration of LPS into plasma leading to alleviating systemic inflammation [47]. Therefore, the findings of this study support the potential for Miang associated synbiotic along with *L. pentosus* A14-6 combined with XOS and synbiotic *L. pentosus* CMY46 combined with GOS to serve as novel synbiotics for protecting colonic inflammation. 

## 5. Conclusions

The findings of this study investigated the treatment with probiotics and synbiotics isolated from Miang ameliorated colonic inflammation induced by DSS through improving mucosal barrier function and regulating the inflammatory response. Moreover, synbiotics showed more effectiveness than probiotics alone or sulfasalazine in protecting DSS-induced colitis. Our study establishes a rationale for synbiotics from Miang as a therapeutic approach in the prevention of UC.

## Figures and Tables

**Figure 1 nutrients-14-00227-f001:**
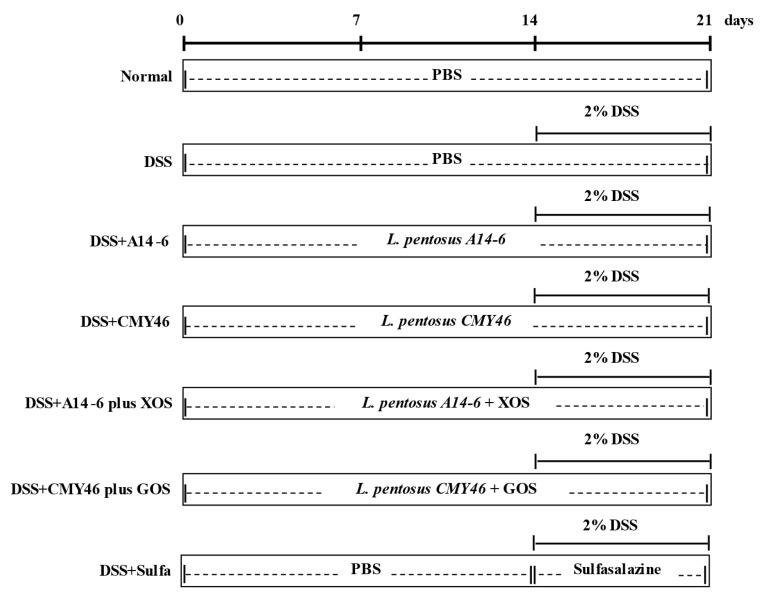
Schematic experimental protocol for investigating the protective effect of probiotics and synbiotics from Miang on experimental colitis in mice. DSS, dextran sulfate sodium; XOS, xylooligosaccharides; GOS, galactooligosaccharides; sulfa, sulfasalazine. PBS, phosphate-buffered saline.

**Figure 2 nutrients-14-00227-f002:**
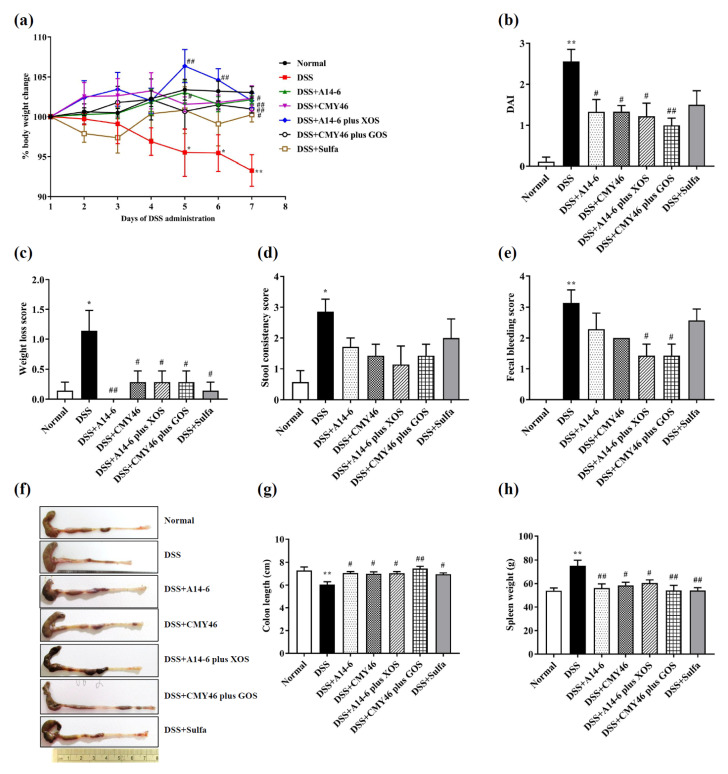
The effects of probiotics and synbiotics on clinical symptoms and severity of colitis induced by DSS. (**a**) The body weight; (**b**) DAI; (**c**) weight loss score; (**d**) stool consistency score; (**e**) fecal bleeding score; (**f**) representative macroscopic images of colons from each mice group; (**g**) colon length; (**h**) spleen weight. Data expressed as mean ± SEM (*n* = 7). * *p* < 0.05; ** *p* < 0.01 vs. normal group. ^#^
*p* < 0.05; ^##^
*p* < 0.01 vs. DSS group.

**Figure 3 nutrients-14-00227-f003:**
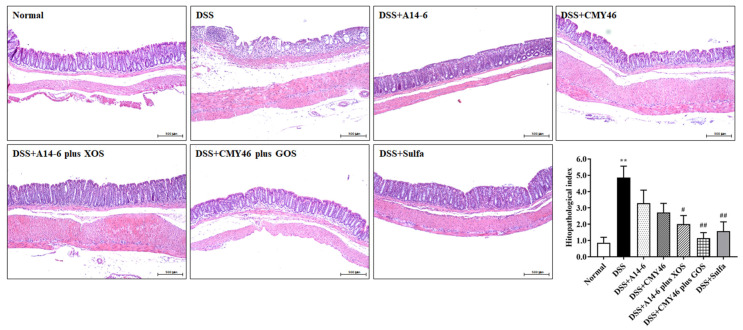
The effect of probiotics and synbiotics on DSS-induced histopathological damage in the colon. Histological images of distal colonic tissues stained with H and E in each experimental group and histopathological index was analyzed under light microscopy. The samples were observed at ×100 magnification. Data expressed as mean ± SEM (*n* = 7), ** *p* < 0.01 vs. normal group. ^#^
*p* < 0.05; ^##^
*p* < 0.01 vs. DSS group.

**Figure 4 nutrients-14-00227-f004:**
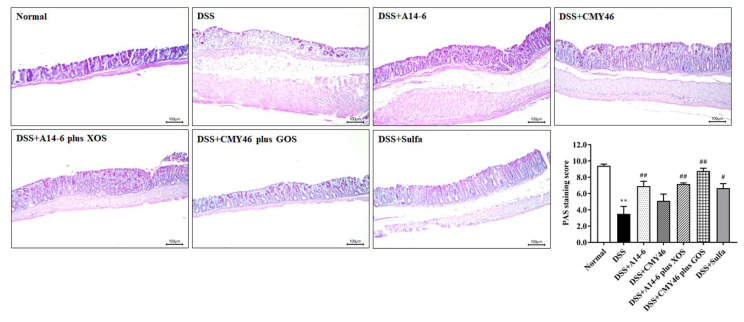
The effect of probiotics and synbiotics from Miang on mucus-secreting goblet cells. PAS staining of mucosal glycoproteins in goblet cells in the apical epithelial cells was observed with intense magenta color in the apical epithelial cells. PAS staining score was analyzed using light microscopy (×200 magnification) from at least five fields. Data are expressed as mean ± SEM (*n* = 7), ** *p* < 0.01 vs. normal group. ^#^
*p* < 0.05; ^##^
*p* < 0.01 vs. DSS group.

**Figure 5 nutrients-14-00227-f005:**
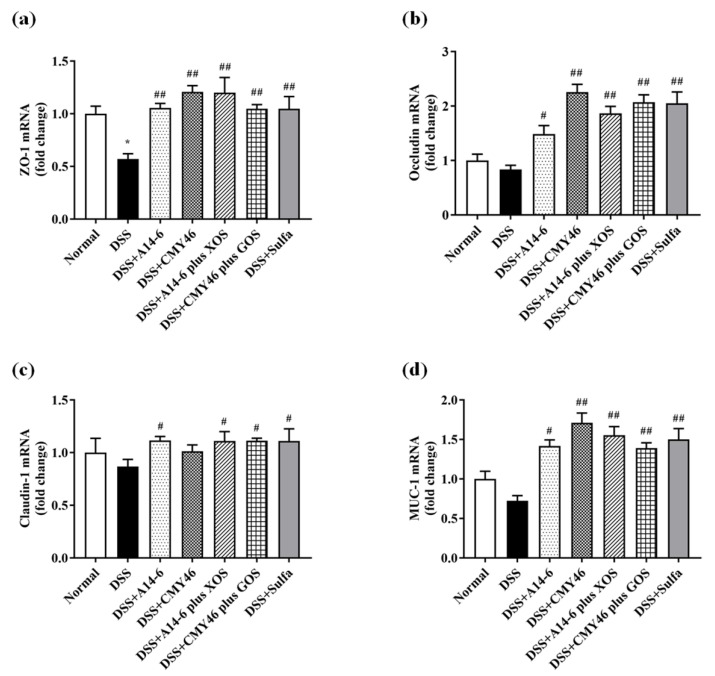
The effect of probiotics and synbiotics from Miang on colonic barrier integrity. (**a**) The expression of ZO-1; (**b**) occludin; (**c**) claudin-1; and (**d**) MUC-1 in the colon were detected by qRT-PCR analysis. Data are expressed as mean ± SEM (*n* = 7), * *p* < 0.05 vs. normal group. ^#^
*p* < 0.05; ^##^
*p* < 0.01 vs. DSS group.

**Figure 6 nutrients-14-00227-f006:**
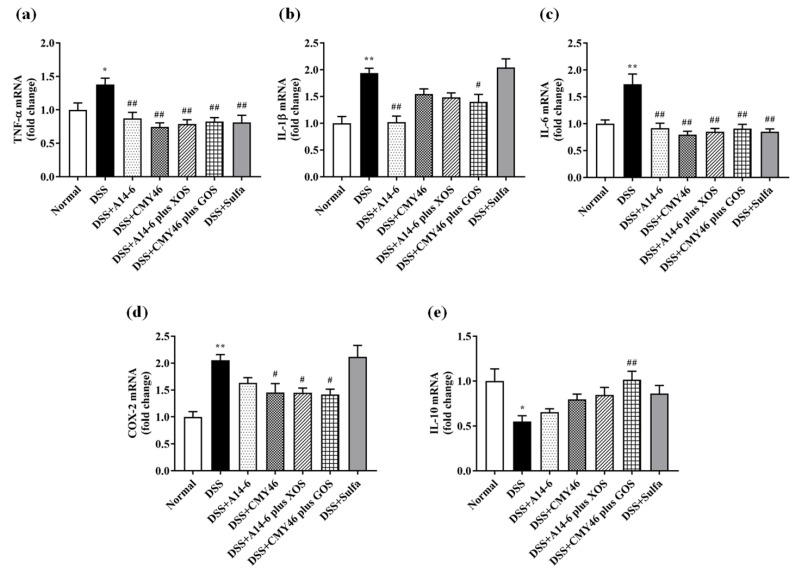
The effect of probiotics and synbiotics from Miang suppressed DSS-induced colonic inflammation. The expression levels of (**a**)TNF-α; (**b**) IL-1β; (**c**) IL-6; (**d**) COX-2; and (**e**) IL-10. Data expressed as mean ± SEM (*n* = 7), * *p* < 0.05; ** *p* < 0.01 vs. normal group. ^#^
*p* < 0.05; ^##^
*p* < 0.01 vs. DSS group.

**Figure 7 nutrients-14-00227-f007:**
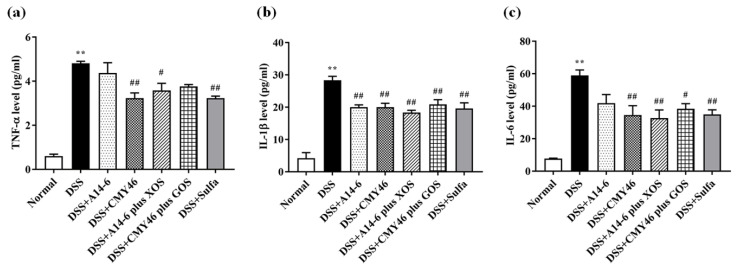
The effects of probiotics and synbiotics from Miang on DSS-induced systemic inflammation. The serum levels of (**a**)TNF-α; (**b**) IL-1β; (**c**) IL-6. Data are expressed as mean ± SEM (*n* = 4), ** *p* < 0.01 vs. normal group. ^#^
*p* < 0.05; ^##^
*p* < 0.01 vs. DSS group.

**Figure 8 nutrients-14-00227-f008:**
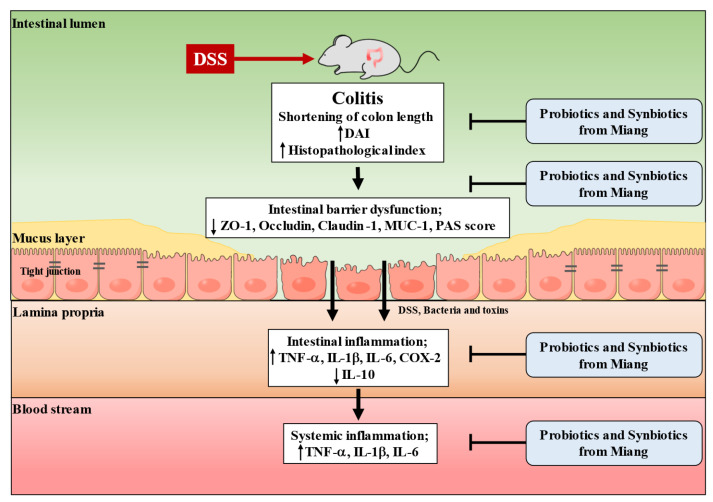
The proposed model represents the protective effect of probiotics and synbiotics on ameliorating intestinal inflammation induced by DSS in mice. DSS, dextran sulfate sodium; DAI, disease activity index; ZO-1, zonula occludin-1; MUC-1, mucin 1; PAS, periodic acid Schiff; TNF- α, tumor necrosis factor-α; IL-1β, interleukin 1β; IL-6, interleukin 6; IL-10, interleukin 10; COX-2, cyclooxygenase 2.

**Table 1 nutrients-14-00227-t001:** Primer sequences used in this study.

Gene	Primer Sequences (5′ to 3′)	
TNF-α	Forward	CGG GCA GGT CTA CTT TGG AG
	Reverse	ACC CTG AGC CAT AAT CCC CT
IL-1β	Forward	AAA AAA GCC TCG TGC TGT CG
	Reverse	GTC GTT GCT TGG TTC TCC TTG
IL-6	Forward	ATC CAG TTG CCT TCT TGG GAC TGA
	Reverse	TAA GCG TCC GAC TTG TGA AGT GGT
IL-10	Forward	TAC CTG GTA GAA GTG ATG CC
	Reverse	CAT CAT GTA TGC TTC TAT GC
COX-2	Forward	TGA GCA CAG GAT TTG ACC AG
	Reverse	CCT TGA AGT GGG TCA GGA TG
ZO-1	Forward	TGG AAT TGC AAT CTC TGG TG
	Reverse	CTG GCC CTC CTT TTA ACA CA
Occludin	Forward	GCT GTG ATG TGT GTG AGC TG
	Reverse	GAC GGT CTA CCT GGA GGA AC
Claudin-1	Forward	TCT ACG AGG GAC TGT GGA TG
	Reverse	TCA GAT TCA GCA AGG AGT CG
MUC-1	Forward	CTG TTC ACC ACC ACC ATG AC
	Reverse	CTT GGA AGG GCA AGA AAA CC
GAPDH	Forward	CAC TCA CGG CAA ATT CAA CGG CAC
	Reverse	GAC TCC ACG ACA TAC TCA GCA C

## Data Availability

Not applicable.

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
