# Peer review of "Protective Effect of Probiotics Isolated from Traditional Fermented Tea Leaves (Miang) from Northern Thailand and Role of Synbiotics in Ameliorating Experimental Ulcerative Colitis in Mice"

_nutrients, 2022, doi:10.3390/nu14010227_

Round 1
Reviewer 1 Report
The aim of this manuscript was to investigate the putative effects of probiotic and synbiotic treatments isolated from thai fermented tea leaves (Miang) in an animal model of ulcerative colitis induced by pharmacological exposure to DSS. The authors found that in general, probiotic treatments exhibited higher beneficial effects than sulfasalazine (control, anti-inflammatory treatment) in alleviating histopathological symptoms and DAI index. Remarkably, synbiotic treatments, in particular the association of CMY46 and GOS, outperformed the effects of single probiotic treatments in DSS-administered animals. While this body of work is substantial and informative, there are several outstanding issues outlined below.
Major issues:
- In the introduction, the authors state that “There are 86 various bioactive compounds in Miang relevant to antioxidant and antimicrobial activities 87 [22]. Previously, L. plantarum A9-2, L. pentosus A14-6, L. pentosus A26-8, L. pentosus CMY46, 88 and Pediococcus pentosaceus CMY46 have been isolated from Miang samples collected from 89 northern Thailand [23]” (lines 86-90). If at least five distinct probiotics have been isolated from Miang, how do the authors justify the selection of certain individual probiotics (A14-6, and CMY46) for their experimental design? Furthermore, how did they decide to associate A14-6 with XOS rather than GOS (and vice versa for CMY46)?
- Additional groups should be added to the experimental design to solidify the original hypothesis. Moreover, as Sulfasalazine administration is associated with demonstrated side effects in clinical populations, several additional groups, including a control group (no DSS – Sulfasalazine alone), could be added to the experimental design.
- The working model proposed in the Discussion section is oversimplistic and does not provide much novelty to the field. Although biochemical and histopathological measurements do point to a decrease of pro-inflammatory markers (TNF-Alpha, IL-1beta, IL-6, COX-2) and parallel restoration of an anti-inflammatory marker (IL-10) in groups treated with pro/synbiotics, those markers are first in line and effectors of downstream inflammatory pathways have not been fully characterized in the present study.
Minor issues:
- Global comment: Typographical errors are present in the text and need correction (example line 90: “In addition”). Sentences are sometimes circumvoluted. For instance, the sentence (lines 316-318) could be reduced by half and made more explicit: Our findings show that the prolonged treatment of probiotics and synbiotics isolated from Miang in mice does not cause mortality.
- M&M section: It would be worth adding further details with respect to the isolation of individual probiotic species from Miang tea leaves. This will help the reader to better capture the involved, but unlisted techniques preceding the context of the described study.
- M&M section: The authors should consider justifying their choice of the concentration of probiotics (line 144) and DSS (line 145), along with the concentration of Sulfasalazine (line 147).
- Results section (Figure 2B): It would be great if the different measures summarized as the hybrid DAI measurement could be displayed individually. Do DSS-treated animals differ from control mice because of stool loss of consistency, presence of blood in feces, or weight loss only?
- Results section (Figure 2C): There are 7 pictures for 7 groups, but the group labels do not match, along with the font. Please advise.
- Results section (Figures 3 and 4): the value of the scale is missing in the caption for these two figures. Please amend accordingly.
Reviewer 2 Report
General comments
The manuscript by Kangwan et al. aims to evaluate the protective effect of probiotics L. pentosus A14-6 and CMY46, and synbiotics L. pentosus A14-6 in combination with XOS and L. pentosus CMY46 in combination with GOS, on dextran sulfate sodium-induced colitis in mice.
The manuscript is very well written and readable. The experimental set-up was clearly presented and the results are fair and interesting.
I would recommend its acceptance after minor revision, according to following concerns.
1) Authors should clarify the rationale of preparing and testing synbiotics L. pentosus A14-6 in combination with XOS and L. pentosus CMY46 in combination with GOS, as there are other possible combinations, i.e. L. pentosus A14-6 in combination with GOS and and L. pentosus CMY46 in combination with XOS. Why they have not been considered?
2) In Figure 6, there are two levels of significance against normal group (* and **), while only one is indicated in the caption.
3) In Figure 7, there is one level of significance against normal group (**), while there is explanation for the other one (*) in the caption.
Round 2
Reviewer 1 Report
Most of the comments have been correctly addressed.